# Cannabidiol changes P-gp and BCRP expression in trophoblast cell lines

Valeria Feinshtein[1], Offer Erez[2], Zvi Ben-Zvi[1], Noam Erez[1],
Tamar Eshkoli[2], Boaz Sheizaf[2], Eyal Sheiner[2], Mahmud Huleihel[3] and
Gershon Holcberg[2]

[1] Department of Clinical Biochemistry and Pharmacology, Ben-Gurion University of the Negev,
Israel
[2] Department of Obstetrics and Gynecology, Soroka University Medical Center, School of
Medicine, Faculty of Health Sciences, Ben-Gurion University of the Negev, Israel
[3] The Shraga Segal Department of Microbiology and Immunology, Ben-Gurion University of the
Negev, Israel

## ABSTRACT

**Objectives.** Marijuana is the most commonly used illicit drug during pregnancy. Due to high lipophilicity, cannabinoids can easily penetrate physiological barriers like the human placenta and jeopardize the developing fetus. We evaluated the impact of cannabidiol (CBD), a major non-psychoactive cannabinoid, on P-glycoprotein (P-gp) and Breast Cancer Resistance Protein (BCRP) expression, and P-gp function in a placental model, BeWo and Jar choriocarcinoma cell lines (using P-gp induced MCF7 cells (MCF7/P-gp) for comparison).

**Study design.** Following the establishment of the basal expression of these transporters in the membrane fraction of all three cell lines, P-gp and BCRP protein and mRNA levels were determined following chronic (24–72 h) exposure to CBD, by Western Blot and qPCR. CBD impact on P-gp efflux function was examined by uptake of specific P-gp fluorescent substrates (calcein-AM, DiOC2(3) and rhodamine123(rh123)). Cyclosporine A (CsA) served as a positive control.

**Results.** Chronic exposure to CBD resulted in significant changes in the protein and mRNA levels of both transporters. While P-gp was down-regulated, BCRP levels were up-regulated in the choriocarcinoma cell lines. CBD had a remarkably different influence on P-gp and BCRP expression in MCF7/P-gp cells, demonstrating that these are cell type specific effects. P-gp dependent efflux (of calcein, DiOC2(3) and rh123) was inhibited upon short-term exposure to CBD.

**Conclusions.** Our study shows that CBD might alter P-gp and BCRP expression in the human placenta, and inhibit P-gp efflux function. We conclude that marijuana use during pregnancy may reduce placental protective functions and change its morphological and physiological characteristics.

## INTRODUCTION

Until recently it was assumed that prenatal exposure to marijuana is frequently combined with other drugs (e.g., tobacco and alcohol), making cannabis effects difficult to isolate

Corresponding author
Valeria Feinshtein,
shteiman@bgu.ac.il

and assess (*Kozer & Koren, 2001*; *Moore et al., 2010*). However, the studies of Dekker et al. and Hayatbakhsh et al. suggested that cannabis use prior to, during the first trimester, or throughout gestation is associated with a higher risk for a low birth weight and neonatal length, as well as preterm labor (*Dekker et al., 2012*; *Hayatbakhsh et al., 2012*). Moreover, the presence of the endocannabinoid receptors CB1 and CB2 on placental syncytiotrophoblast (*Habayeb et al., 2008a*), along with marijuana being the most popular drug of abuse among pregnant population (*Brown & Graves, 2013*), raised the need to understand the effect of cannabinoids on the placenta.

Cannabidiol (CBD) is one of the most abundant cannabinoids in the marijuana plant (*Mechoulam & Shvo, 1963*; *Mechoulam & Hanuš, 2002*; *Schier et al., 2012*). It is a promising candidate for clinical utilization, due to low affinity binding to CB1 and CB2 cannabinoid receptors and no cognitive and psychoactive activity (*Zuardi, 2008*; *Deiana, 2012*).

Preliminary data from *in vitro* models indicate that cannabinoids may interact with human P-gp (ABCB1) and BCRP (ABCG2). Acute and long-term exposures to cannabinoids were shown to alter P-gp expression and function (*Holland et al., 2006*; *Zhu et al., 2006*). However, BCRP exposure to cannabinoids demonstrated functional inhibition only (*Holland et al., 2007*).

P-gp and BCRP are both thought to be protective for the fetus. These ATP-binding cassette (ABC) efflux transporters expressed at the apical membrane of the polarized syncytiotrophoblast layer (*Ni & Mao, 2011*), and play a significant role in drug transfer across the placental barrier (*Lankas et al., 1998*; *Mao, 2008*; *Vahakangas & Myllynen, 2009*; *Myllynen, Kummu & Sieppi, 2010*; *Eshkoli et al., 2011*). P-gp was found to be an anti-apoptotic cellular agent (*Smyth et al., 1998*; *Huls, Russel & Masereeuw, 2009*) and BCRP was shown to have a role in placental tissue and syncytial survival by protecting cells from pro-apoptotic injuries (*Evseenko et al., 2007*; *Evseenko, Paxton & Keelan, 2007*; *Hardwick, Velamakanni & van Veen, 2007*; *Vahakangas & Myllynen, 2009*).

We have already assessed the interaction of CBD with placental BCRP, finding that in both, *in vitro* and *ex vivo* systems, CBD inhibited its efflux function (*Feinshtein et al., 2013*). These results together with other recent findings regarding the effect of cannabinoid on the ABC transporter led us to investigate whether CBD affects placental P-gp on functional and placental P-gp and BCRP on expressional levels (*Holland et al., 2006*; *Zhu et al., 2006*; *Holland et al., 2007*; *Holland, Allen & Arnold, 2008*).

In the present work the implications of CBD exposure on P-gp and BCRP expression and P-gp function is tested in a human trophoblast-like cell lines BeWo and Jar, as placental model.

## MATERIALS AND METHODS

### Materials

BeWo and Jar cells were obtained from Dr. B. Ugele, Ludwig-Maximilians University, Munich, Germany (*Feinshtein et al., 2010*; *Polachek et al., 2010*; *Feinshtein et al., 2013*). MCF7/P-gp cells (BCRP expressing and P-gp induced cells), were kindly provided by Prof. Esther Priel (Ben Gurion University, Beer Sheva, Israel) (*Feinshtein et al., 2013*). All

materials for cell culture were purchased from Biological Industries (Israel). CBD was a kind gift from Prof. Raphael Mechoulam (The Hebrew University of Jerusalem, Jerusalem, Israel).Calcein-AM, DiOC2(3), rh123 and cyclosporine A (CsA) were purchased from Sigma-Aldrich (St. Louis, MO, USA). A list of all antibodies used in the current research is summarized in Table S1.

## METHODS

### Cell culture and drug treatments

MCF7/P-gp cells were cultured in conditions as previously described (*Golan, Schreiber & Avissar, 2009*), BeWo and Jar cells were cultured as previously described (*Golan, Schreiber & Avissar, 2009*; *Feinshtein et al., 2010*). Briefly, 24 h after seeding in $35 \times 10$ mm cell culture dishes (Corning), growth medium (DMEM for MCF7/P-gp, DMEM/F-12 for BeWo and Jar cells, supplemented with 10% fetal bovine serum, 2 mM L-glutamine, 100 μg/ml strep-tomycin and 100 units/ml penicillin in a humidified atmosphere of 95% air and 5% $CO_2$ at 37°C) was replaced with fresh medium containing CBD 10 or 15 μM (initially dissolved in DMSO) or DMSO (0.08% or 0.12%, respectively, vehicle control). In order to achieve long-term exposure, the treatment medium was refreshed every day for 24 h, 48 h or 72 h.

### Subcellular fractionation

Fractionation procedure was carried out as previously described (*Golan, Schreiber & Avissar, 2009*), for the isolation of the membrane fraction. The membrane fraction was diluted in sample buffer 1:3 (10% v/v glycerol, 20% v/v SDS 20%, 5% v/v $\beta$-mercaptoethanol, 0.05% w/v bromophenol blue, pH 6.8), boiled for 5 min at 95°C (for BCRP determination) or incubated for 30 min at 37°C (for P-gp determination) and frozen at −80°C until assayed. Aliquots were taken for protein determination using the Lowry assay. Fractions purity was verified using specific markers: $Na^+/K^+$ ATPase for membranes and NF/kB p65 for cytosol.

## IMMUNOCYTOCHEMISTRY AND CONFOCAL MICROSCOPY

MCF7/P-gp, BeWo and Jar cells were seeded on cover slips and grown to 70% sub-confluency. Cells were then fixed in PFA for 15 min and stained with mouse anti-human CD 243 (MDR-1) antibodies, diluted in PBS, containing 3% BSA, for 1 h at room temperature, followed by incubation with Alexa Fluor 488 goat anti-mouse antibody in PBS containing 3% BSA, for 1/2 h at room temperature. Cover-slips were washed 3 times and mounted onto glass slides with DAPI (4′,6-diamidino-2-phenylindole)-containing fluorescent mounting medium (DAPIFluoromount-G; SouthernBiotech). Immunofluo-rescence was detected by an Olympus FV-1000 Spectral confocal laserscanning microscope with excitation at 488 nm and emission at 520 nm. Image analysis was performed using ImageJ v. 1.40C software.

## FACS analysis of BCRP levels

Cells were grown to confluency in 6-well culture dishes, trypsinized, counted, and $0.5 \times 10^6$ cells were placed in light resistant vials. Cells were washed three times with cold PBS (and each time centrifuged at 1200 g for 5 min), and incubated for 30 min with BSA 3% in PBS. Following additional PBS wash, 50 µL of BSA 1% in PBS and antibodies were added (1:10 anti-BCRP FITC conjugated antibody or 1:10 CBL602F FTIC conjugated isotype negative control). Cells were incubated in the dark at 4°C for 1 h, washed with ice cold PBS, re-suspended (in 0.5 ml PBS) and analyzed by FACS (BD FACS Vantage).

## Immunoblotting

Membrane fractions were thawed on the day of assay. Protein aliquots (60–100 µg/lane) were taken for protein separation by SDS-PAGE as previously described (*Golan, Schreiber & Avissar, 2009*). Semi-quantitative analysis was carried out using a computerized image analysis system (EZQuant-Gel 2.11; EZQuant Biology Software Solutions Ltd., Israel). Equal protein loading was ensured by normalization to $Na^+/K^+$ ATPase (for P-gp) or actin (for BCRP).

## RNA extraction, reverse transcription, real-time polymerase chain reaction (qPCR)

Isolation and purification of total RNA from BeWo and Jar cells was carried out using EZ-RNA Kit (Biological Industries, Israel) according to manufacturer's instructions. 1 µg of total RNA was used for reverse transcription using High Capacity cDNA RT kit (Applied Biosystems, Foster City, CA), in 20 µl reaction volume. BCRP and P-gp mRNA was measured by qRT-PCR, as indicated in the manufacturer protocol (Applied Biosystems, Foster City, CA), and performed by the Applied Biosystems Real Time PCR system (7500 system), using TaqMan probes and primers for human BCRP, P-gp and actin (Applied Biosystems, Foster City, CA). The cycling conditions for all primers were as follows: hold for 10 min at 95°C, followed by 40 cycles consisting of two steps, 15 s at 95°C (denaturing), and 1 min at 60°C (annealing-extension). The threshold cycle, which correlates inversely with the mRNA levels of target, was measured as the cycle number at which the reporter fluorescent emission increases above a threshold level. P-gp and BCRP mRNA levels were normalized to actin mRNA in the same samples. Results were analyzed by the $2^{-\Delta-\Delta C_T}$ method, demonstrating the relative changes in gene expression from real-time quantitative PCR experiments, using 7500 Software v2.0.4 (Applied Biosystems, Foster City, CA).

## P-gp substrate uptake experiments

Cells were pre-incubated for 30 min with CBD 10 or 25 µM (working concentration previously published (*Holland et al., 2006*; *Ligresti et al., 2006*; *Zhu et al., 2006*; *Holland et al., 2007*; *Holland, Allen & Arnold, 2008*; *De Filippis et al., 2011*; *Arnold et al., 2012*; *Harvey et al., 2012*; *Hill et al., 2012*; *Maor et al., 2012*; *Solinas et al., 2012*; *Dudášová et al., 2013*; *Juknat et al., 2013*; *Nabissi et al., 2013*), and initially dissolved in DMSO) or CsA 20 µM (known P-gp inhibitor used as positive control (*Mori et al., 2012*)), dissolved in transport buffer (TB) (pH = 7.4) (*Feinshtein et al., 2010*), while "control cells" were pre-incubated

in TB with the correlating concentration of DMSO. Following pre-incubation, P-gp substrates were added (Calcein-AM or DiOC2(3) or rh123) (*Minderman et al., 1996*; *Martin et al., 2003*) and cells were further incubated for 30 min. At the end of incubation plates were treated as previously detailed (*Feinshtein et al., 2010*), and samples were stored (at −20°C) for further analysis. Intracellular fluorescence of all P-gp substrates was quantified by Infinite M200 microplate reader (Tecan) and normalized to protein amount (determined by Lowry method) (*Lowry et al., 1951*), or detected by an Olympus FV-1000 Spectral confocal laserscanning microscope and analyzed using ImageJ v. 1.40C software.

### Statistical analysis

All statistics and graphs were carried out using GraphPad Prism5 software. Student's *t*-test or one-way ANOVA followed by appropriate Bonferroni corrections were used.

## RESULTS

### CBD impact on BCRP and P-gp protein expression

The changes in P-gp and BCRP protein levels in the membrane fraction of BeWo, Jar and MCF7/Pgp cells were studied. Following determination of proper cell fractionation (Fig. 1A), the basal expression of these two transporters in non-treated cells was determined by Western Blot analysis (Fig. 1B). Due to unexpected P-gp expression profile in all three cell lines we verified its basal expression by immunocytochemical fluorescent staining (Fig. 1C). It can be seen that our BeWo, Jar and MCF7/P-gp cells express detectable levels of P-gp. For P-gp expression, the results of the Western Blot were confirmed by immunocytochemistry showing that P-gp expression was the highest in MCF7/P-gp and the lowest in Jar cells. For BCRP (ABCG2), results were confirmed by FACS analysis, showing that BCRP expression was much higher in JAR cells compared to BeWo cells (in full accordance with Western Blot analysis) (Fig. 1D).

The changes in membrane BCRP and P-gp levels in BeWo cells following long-term exposure to CBD are displayed in Fig. 2. Indeed, BCRP levels were significantly increased in a concentration-dependent manner (Fig. 2B) that was not time-dependent (Fig. 2A). Following long-term exposure to CBD, P-gp protein levels significantly decreased, in a concentration-dependent manner (Fig. 2D), with no time-dependent effect (Fig. 2C). Similarly, in Jar cells (Fig. 3), BCRP protein concentrations was significantly elevated following long term exposure to CBD, in a concentration but not time-dependent effect (Figs. 3A and 3B). Due to very low P-gp initial (baseline) expression in the Jar cell line (Figs. 1B and 1C), the expected down-regulation in P-gp expression following long-term exposure to CBD is demonstrated in experiments of 72 h treatment only. P-gp is vaguely visualized in the control group while it is almost undetectable in the CBD treated group (Fig. 3C). 48 h exposure yielded the same results (P-gp levels were almost undetectable – data not shown), making time and concentration dependent comparisons impractical.

To examine whether CBD effects are cell specific, the same experimental routine was applied to MCF/P-gp cells. In these cells, BCRP and P-gp behavior under CBD

 

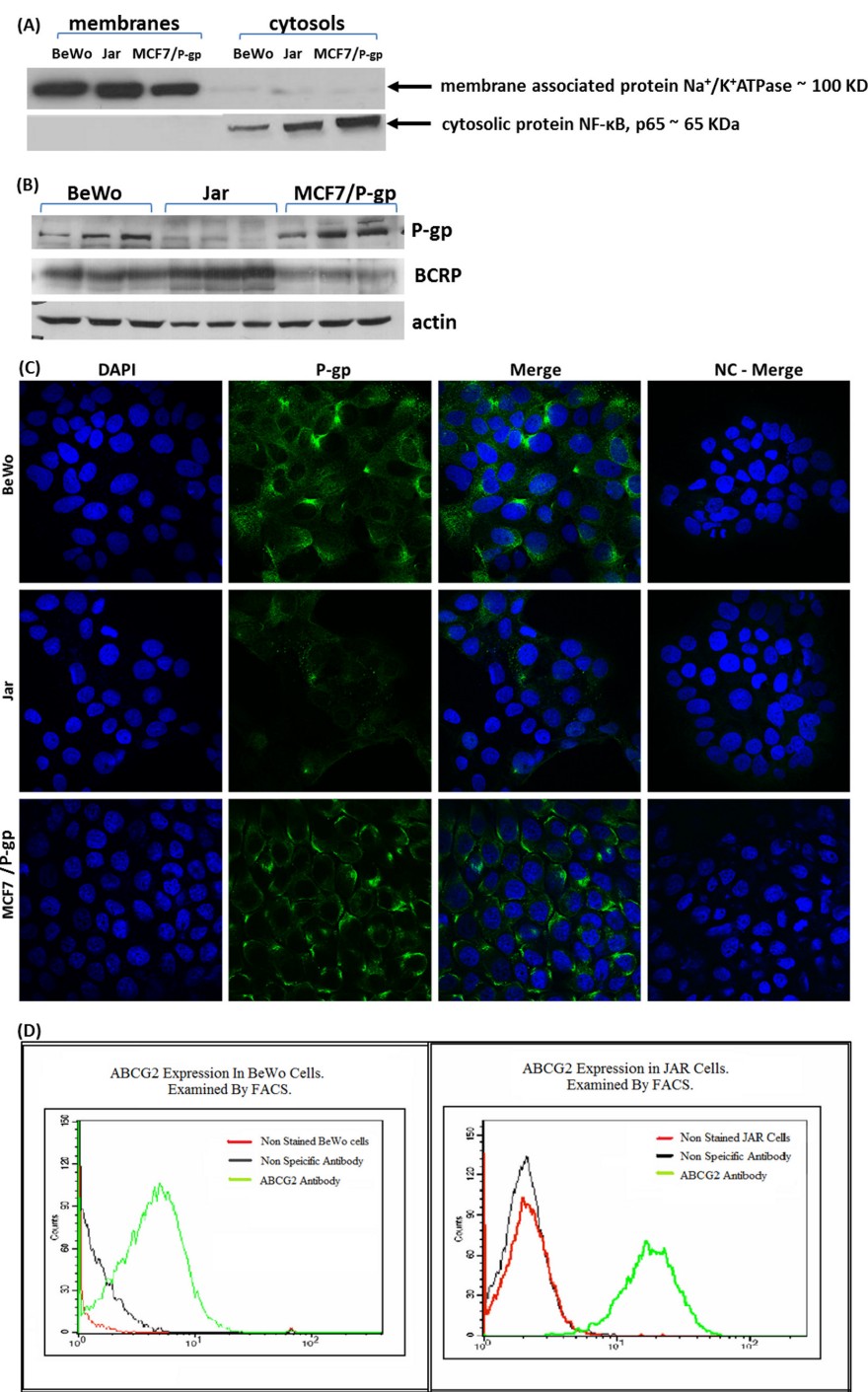

**Figure 1 Cell fractionation, P-gp and BCRP basal expression.** (A) Cell fractionation: to verify proper fractionation of whole cell lysate, membrane fractions and cytosolic fractions of BeWo, Jar, MCF/P-gp cells were subjected to Western Blot analysis. $Na^+/K^+$ ATPase served as membrane marker, and NF-$\kappa$B (p65 subunit) served as cytosolic marker. (B) P-gp and BCRP basal expression in MCF7/P-gp, Jar and BeWo cells was determined by Western Blotting. (C) Immunocytochemistry: fluorescent staining of BeWo, Jar and MCF7/P-gp cells with anti-P-gp antibody and DAPI. NC – negative control. (D) BCRP (ABCG2) expression in choriocarcinoma cell lines BeWo and Jar as demonstrated by FACS analysis.

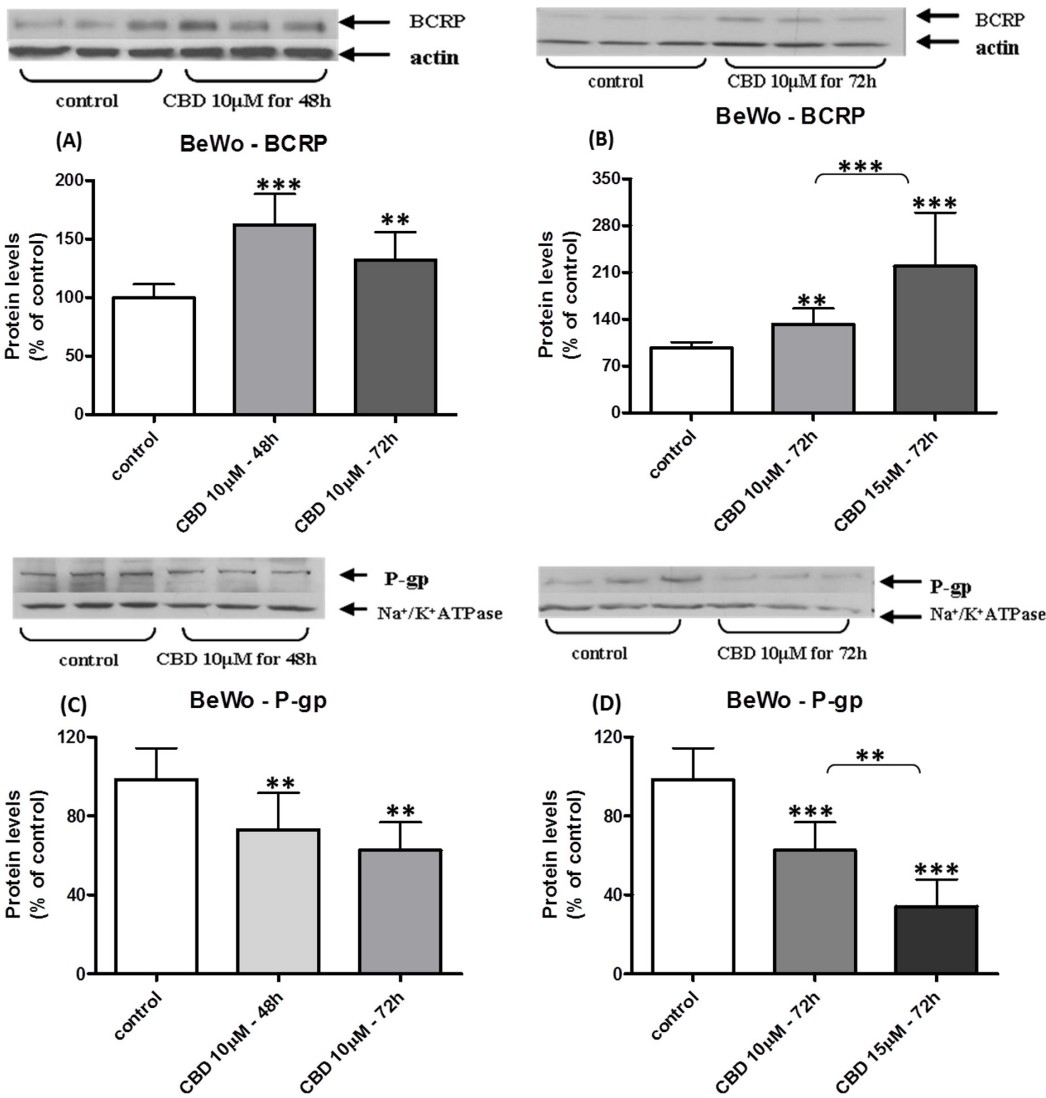

**Figure 2 Long-term exposure of BeWo cells to CBD: changes in P-gp and BCRP protein levels.** Long-term exposure of BeWo cells to CBD: changes in membrane BCRP and P-gp expressed levels. Changes in BCRP-CBD concentration-dependent (B) and time-dependent (A). Changes in P-gp-CBD concentration-dependent (D) and time-dependent (C). No statistical significance between 48 and 72 h time points. Data is displayed in means ± s.d. of at least three ($n = 9$) independent experiments of each concentration or time point. One representative blot is presented for each experimental group. One-Way ANOVA, followed by Bonferroni's multiple comparison test, **$p < 0.01$, ***$p < 0.0001$ compared to control or comparison between groups as indicated.

influence was profoundly different from that seen in choriocarcinoma cell lines. P-gp levels dramatically increased following long-term treatment with CBD, in a concentration- but not time-dependent manner (Fig. 4A). At the same time, BCRP expression was not affected by CBD (neither time-, nor concentration-dependent effect was observed) (Fig. 4B). Thus, we conclude that CBD has a cell-type specific influence upon long-term cellular exposure. CBD impact on MCF7/P-gp cells was not investigated further.

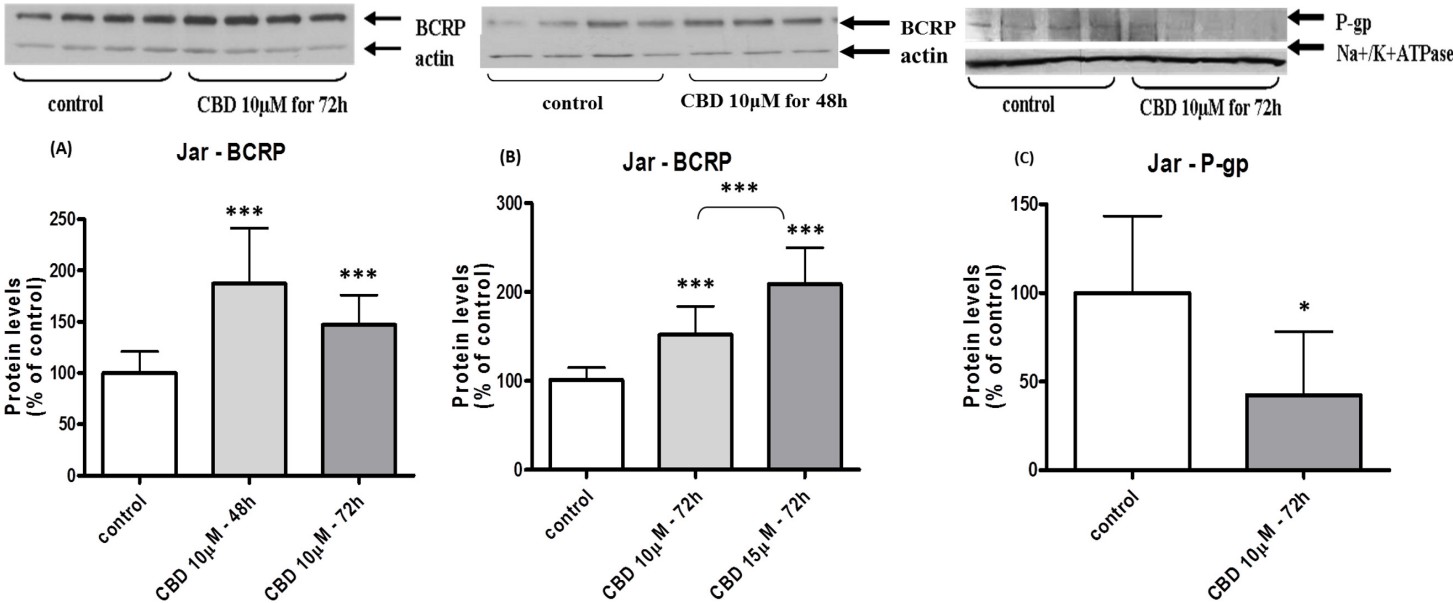

**Figure 3 Long-term exposure of Jar cells to CBD: changes in P-gp and BCRP protein levels.** Long-term exposure of Jar cells to CBD: changes in membrane BCRP and P-gp expressed levels. Changes in BCRP-CBD concentration-dependent (B) and time-dependent (A). Changes in P-gp (C). Data is displayed in means ± s.d. of at least three ($n = 12$) independent experiments of each concentration or time point. One representative blot is presented for each experimental group. One-Way ANOVA, followed by Bonferroni's multiple comparison test (Student's $t$ test for (C)), *$p < 0.05$, ***$p < 0.0001$ compared to control or comparison between groups as indicated.

## CBD impact on BCRP and P-gp mRNA

To have a glimpse into the mechanism underlying the changes seen on the protein level in BCRP and P-gp, we further focused to track the changes that occur on mRNA level of these two transporters. In both, BeWo and Jar cells, BCRP mRNA quantification showed elevation following long-term exposure to CBD (Figs. 5A and 5C), and matched the findings on protein level, supporting transcriptional up-regulation. Likewise, P-gp mRNA expression significantly dropped following long-term treatment with CBD, matching the outcomes seen on the protein level (Figs. 5B and 5D), indicating that the changes in P-gp levels following long-term CBD exposure results from transcriptional down-regulation rather than post-transcriptional changes.

## CBD impact on P-gp function

Only BeWo and MCF7/P-gp cell lines were tested for P-gp inhibition by CBD due to the fact that Jar cells expressed very low P-gp levels. In a preliminary study we observed a non-cell type specific P-gp inhibition by 25 μM CBD in both cell lines, as intracellular calcein and rh123 fluorescence was significantly higher (elevation of $101 \pm 50\%$ and $70 \pm 30\%$, respectively) in the presence of CBD (Figs. 6A and 6B). We further examined whether lower concentration of CBD 10 μM also inhibits P-gp, using two different P-gp specific substrates, calcein-AM and DiOC2(3). P-gp was inhibited following short-term (1 h) exposure to CBD, as significantly more DiOC2(3) and calcein (elevation of $20 \pm 8\%$ and $24 \pm 9\%$, respectively) accumulated in the cells (Figs. 7A and 7B).

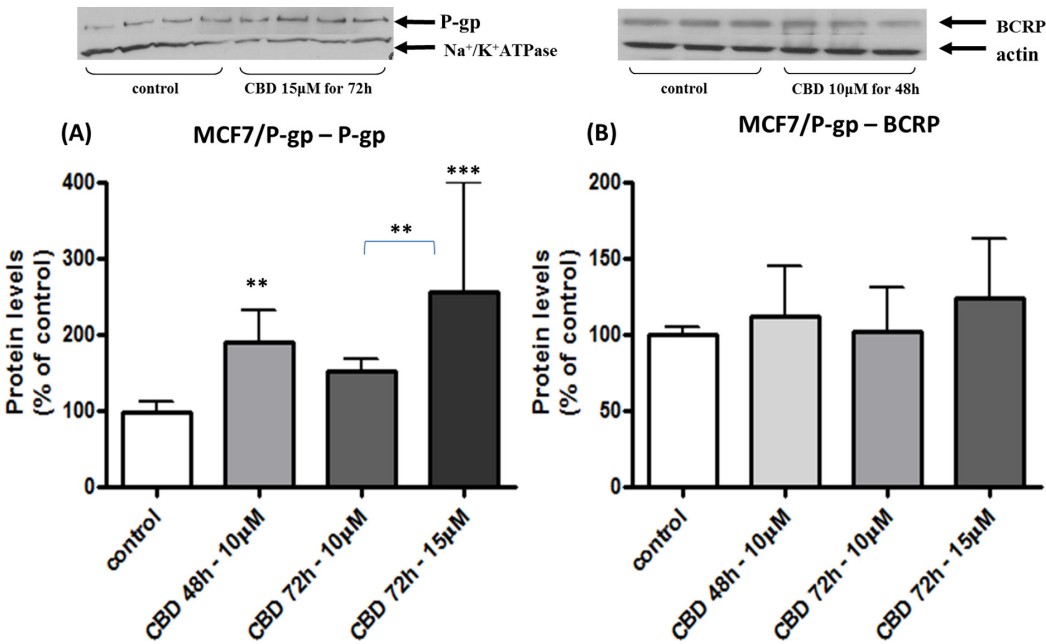

**Figure 4** **Long-term exposure of MCF/P-gp cells to CBD: changes in P-gp and BCRP protein levels.**
Long-term exposure of MCF/P-gp cells to CBD: changes in membrane P-gp (A) and BCRP (B) expressed
levels. Protein levels are given as percent of control levels. Data is displayed in means ± s.d. of at least
three ($n = 9$) independent experiments of each concentration or time point. One representative blot is
presented for each experimental group. One-Way ANOVA, followed by Bonferroni's multiple comparison
test. **$p < 0.01$, ***$p < 0.0001$ compared to control or comparison between groups as indicated.

## DISCUSSION

### Principal findings of the study

There is a dual effect of CBD on the expression of BCRP and P-gp transporters in
trophoblast-like and in MCF7/P-gp cell lines. Under long-term CBD exposure BCRP
and P-gp present cell-type specific changes in protein and mRNA levels. Occurring already
at the transcriptional level, P-gp protein expression is down-regulated, while that of BCRP
is up-regulated. Upon short-term exposure, P-gp efflux function is inhibited by CBD.

### Cannabinoid effects on pregnancy

Cannabis is extensively used in Western society as a recreational drug. The effect of this
drug on pregnancy outcome was under constant debate; however, recently cannabis
consumption was reported to be associated with adverse pregnancy outcomes, including
preterm birth and fetal growth restriction (*Dekker et al., 2012*; *Hayatbakhsh et al., 2012*).
Although the mechanisms in which phytocannabinoids exert their effects are not well
understood, there is accumulating evidence that endocannabinoids (like anandamide)
can influence reproduction, including fertilization, implantation, angiogenesis, embryo
development and placental growth (*Taylor et al., 2007*; *Habayeb et al., 2008a*; *Lewis et al.,
2012*; *Solinas et al., 2012*; *Sun & Dey, 2012*). One possible route to convey the effect of
endocannabinoids is through the CB1 and CB2 receptors that are expressed by the human

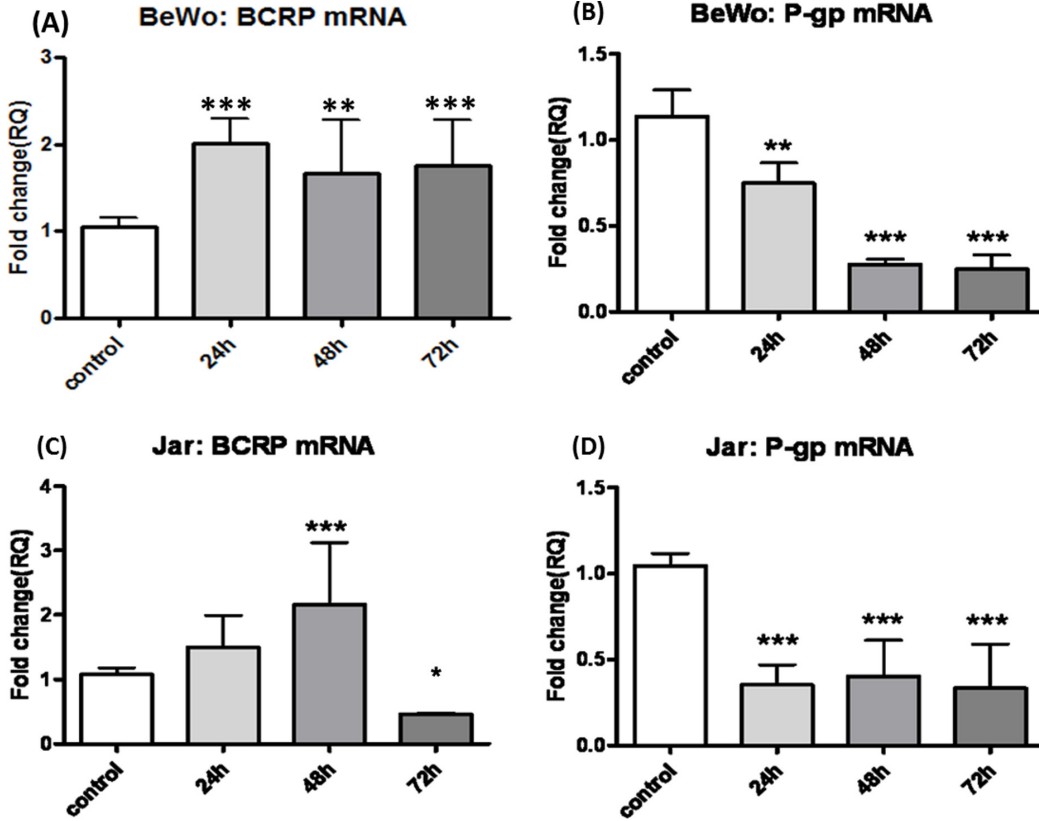

**Figure 5 Long-term exposure of BeWo and Jar cells to CBD: changes in BCRP and P-gp mRNA levels.** (A), (C) changes in BCRP mRNA in BeWo and Jar cells (respectively). (B), (D) changes in P-gp mRNA in BeWo and Jar cells (respectively). Values are given as fold of change compared to control. Data is displayed as means ± s.d. of at least three ($n = 6$) independent experiments for each time point. One-Way ANOVA, followed by Bonferroni's multiple comparison test. *$p < 0.05$, **$p < 0.01$, ***$p < 0.0001$ compared to control.

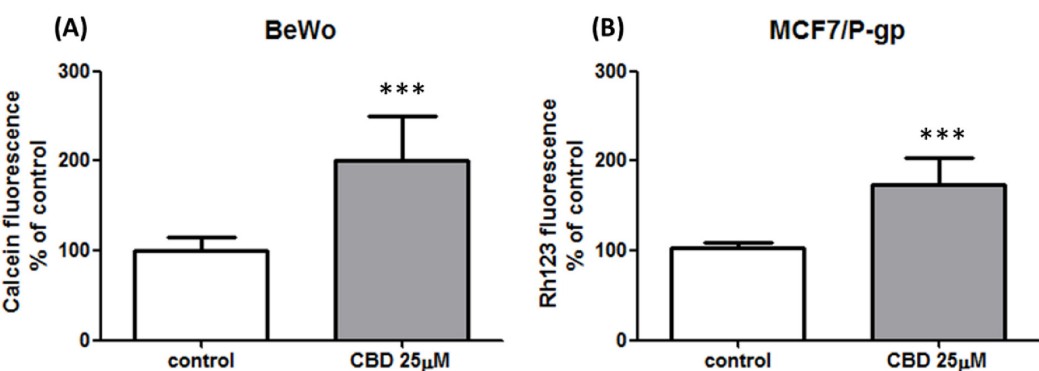

**Figure 6 Non-cell type specific inhibition of P-gp by CBD 25 μM.** Non-cell type specific effect of CBD 25 μM on P-gp dependent calcein and rh123 intracellular efflux in (A) BeWo and (B) MCF7/P-gp cells. Data presented from at least 2 independent experiments ($n = 33, n = 11$, respectively), as means ± s.d. Statistical significance determined by Student's $t$-test. ***$p < 0.0001$.

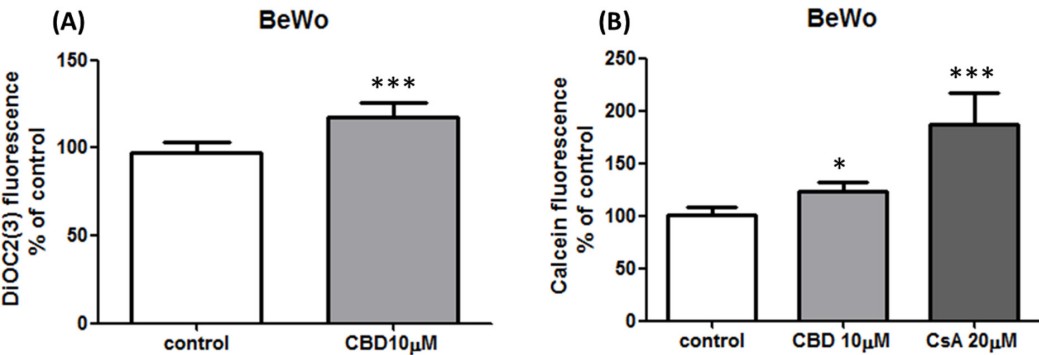

**Figure 7 Inhibition of P-gp by CBD 10 μM in BeWo cells.** CBD 10 μM inhibition of P-gp dependent efflux of calcein (A) and DiOC2(3) (B). Data presented from at least 2 independent experiments ($n = 9$, $n = 6$, respectively), as means ± s.d. Statistical significance determined by One-Way ANOVA, followed by Bonferroni's multiple comparison test for (A) and student's $t$-test for (B). ***$p < 0.0001$, *$p < 0.05$.

trophoblast and provide a direct target for cannabinoids (*Kenney et al., 1999*; *Habayeb et al., 2008a*; *Habayeb et al., 2008b*). However, not all endocannabinoids exert their effect through the classic cannabinoids receptors. For example, CBD lacks the central effects of cannabis and works through CB1 and CB2 independent model of action (*Scuderi et al., 2009*). Since our goal was to test the direct cannabinoid effect on trophoblast transporters BCRP and P-gp, and CBD showed the most potent inhibitory effect (among major cannabinoids) on these transporters (*Holland et al., 2006*; *Zhu et al., 2006*; *Holland et al., 2007*) it made CBD the ideal cannabinoid for the present study.

## BCRP and P-gp significance in pregnancy

BCRP and P-gp play a key role in the transport of drugs and endogenous compounds in the human placenta, affecting the outcome of pregnancy (*Robey et al., 2009*; *Myllynen, Kummu & Sieppi, 2010*). They transport a broad variety of structurally diverse compounds, some of which are congruent (*Frohlich et al., 2004*; *Mathias, Hitti & Unadkat, 2005*; *Zhou, 2008*). In addition, BCRP transports a wide range of substrates, including fetal hormonal precursors such as estrone-3-sulfate, naturally occurring carcinogens, porphyries and ceramides (*Imai et al., 2003*; *Krishnamurthy & Schuetz, 2005*; *Evseenko et al., 2007*; *Evseenko, Paxton & Keelan, 2007*; *Mao, 2008*; *Dietrich et al., 2011*). P-gp is expressed in the apical membrane of syncytiotrophoblast and probably is the main placental protective transporter during the first trimester. Its expression (both mRNA and protein) is the highest during the first trimester and decreases with advancing gestation (*Gil et al., 2005*; *Mathias, Hitti & Unadkat, 2005*; *Sun et al., 2006*). BCRP plays an important role as a survival factor in BeWo cells as well as in the human placenta. It is thought to have a protective antiapoptotic role in the trophoblast, regulating their survival under low oxygen conditions (*Krishnamurthy & Schuetz, 2006*; *Yeboah et al., 2006*; *Evseenko et al., 2007*; *Evseenko, Paxton & Keelan, 2007*).

The trophoblast expression of BCRP may change in different pregnancy complications. Indeed, placentas of women with preterm labor and intra-amniotic inflammation had higher expressions of this transporter than those of women with preterm labor without

inflammation. In addition, the mRNA expression of BCRP correlated with that of IL-8, which also increased significantly in placentas of women with preterm labor and inflammation, suggesting that the transfer of drugs across the placenta may be altered in cases of preterm labor with inflammation (*Mason et al., 2011*).

## CBD impact on BCRP and P-gp upon long-term exposure

In the current study we have demonstrated for the first time that the exposure of trophoblast-like cell lines BeWo and Jar to CBD is associated with two distinct patterns of effects on the expression of the BCRP and P-gp transporters. The use of these cell lines, instead of primary trophoblast cultures, results from the fact that primary trophoblasts may rapidly differentiate in culture, continuously changing their gene and protein expression (*Evseenko, Paxton & Keelan, 2006*). Comparison of these parameters with regard to CBD influences in such a dynamic *in vitro* environment would be almost impossible. Moreover, BeWo and Jar cell lines, derived from human gestational choriocarcinoma, commonly used as an *in vitro* model for trophoblast toxicology studies, and they offer a suitable model to study certain aspects of human trophoblast physiology, without the aspect of inter-patient variability (*Sullivan, 2004*; *Khare et al., 2006*; *Myren et al., 2007*).

The first effect that was observed following long-term exposure is that CBD transcriptionally inhibited P-gp membrane expression (as mRNA levels of P-gp were also dramatically reduced following long-term exposure to CBD). The long-term inhibitory effect of CBD on P-gp protein was previously reported in drug-selected human T lymphoblastoid leukaemia cell line (CEM/VLB(100)) (*Holland et al., 2006*). The cell specificity of CBD was previously reported (*Khare et al., 2006*; *Hu, Ren & Shi, 2011*), yet our study is the first to detect this phenomenon in trophoblast-like cell lines. This observation may have clinical implications, in light of the role of P-gp as one of the key transport mechanisms for numerous drugs in the human placenta, especially during the venerable period of the first trimester in which all the fetal organs are formed. Thus, our results might be especially important to women who use cannabis on a regular basis during the first trimester and are treated with other drugs that are P-gp substrates.

The expressional up-regulation of BCRP mRNA and protein in trophoblast-like cell lines following long-term exposure to CBD is the second novel effect reported in our study. This observation is similar to that reported in trophoblasts of women with preterm labor and inflammation, suggesting that (I) long-term exposure to CBD may elicit an inflammatory response in trophoblast-like cell lines (*Mason et al., 2011*); and (II) that this phenomenon could be a direct compensational consequence of P-gp down-regulation, providing a working defense line to the developing embryo. Interestingly, compensation of such nature was already described in murine placentas (*Hutson, Koren & Matthews, 2010*). Nonetheless, the signal transduction of CBD action in human placenta/trophoblast/choriocarcinoma cells has not been clearly reported yet and needs to be further elucidated.

## CBD impact on P-gp upon acute exposure

In agreement with previously published data, we found that CBD holds inhibiting properties over P-gp efflux function (*Holland et al., 2006*; *Nieri et al., 2006*; *Zhu et al., 2006*). Moreover, our results show inhibition of P-gp efflux function in cell lines that naturally express this transporter. It can be seen that CBD 10 µM (Fig. 7) yielded inhibition effects lower than CBD 25 µM (Fig. 6). Similar to results we recently presented (*Feinshtein et al., 2013*), this observation could imply that CBD inhibits P-gp in a concentration dependent fashion. However, due to different quantification methods used in the present study, this should be further elucidated.

Of note, many of the drugs that are prescribed and considered safe to use during pregnancy are in fact P-gp or BCRP substrates, like Loratadine and $H_2$ blockers (i.e., Ranitidine and Cimetidine) (*Collett et al., 1999*; *Chen et al., 2003*; *Li et al., 2008*; *Schwarz et al., 2008*; *Dahan & Amidon, 2009*; *Gill, O'Brien & Koren, 2009*; *Matok et al., 2010*). Our finding may have clinical implications, suggesting that the use of cannabis during gestation may alter drug transport through the trophoblast and lead to the absence of a functional placental barrier during the first trimester, leaving the developing embryo unprotected at this vulnerable period of pregnancy. Moreover, the trophoblasts of pregnant women exposed to marijuana may exhibit some resistance to apoptotic and inflammatory processes due to the effect of CBD on BCRP expression.

## CONCLUSIONS

Following cannabis consumption, all the drugs that are P-gp substrates can potentially penetrate the human placental barrier at higher rates when combined with CBD, and therefore their safety under these conditions is to be questioned. Additionally, changes in placental BCRP expression profile might lead to altered transplacental transport of BCRP substrates, such as medications, naturally occurring carcinogens, hormonal precursors and apoptotic molecules, and influence pregnancy outcomes.

## ACKNOWLEDGEMENTS

We gratefully thank Prof. Sofia Schreiber-Avissar from the Department of Clinical Biochemistry and Pharmacology, Faculty of Health Sciences, Ben-Gurion University of the Negev, Beer-Sheva, Israel, for providing writing assistance.

### Funding

The authors declare that there was no funding for this work.

### Competing Interests

Offer Erez is an Academic Editor for PeerJ.

## Author Contributions

- Valeria Feinshtein conceived and designed the experiments, performed the experiments, analyzed the data, wrote the paper.
- Offer Erez analyzed the data, wrote the paper.
- Zvi Ben-Zvi conceived and designed the experiments.
- Tamar Eshkoli, Boaz Sheizaf and Mahmud Huleihel contributed reagents/materials/analysis tools.
- Eyal Sheiner analyzed the data, contributed reagents/materials/analysis tools, wrote the paper.
- Gershon Holcberg conceived and designed the experiments, analyzed the data, wrote the paper.

## Supplemental Information

Supplemental information for this article can be found online at http://dx.doi.org/10.7717/peerj.153.

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
