# Peer review of "Cannabidiol changes P-gp and BCRP expression in trophoblast cell lines"

_PeerJ, doi:10.7717/peerj.153_

## Round 0.1 · original submission · Major Revisions

Some major technical issues (PCR, western) were raised by the reviewers. Functional significance of the observed changes need to be proven. Additional experiemnts are required.

Reviewer 1 ·

Basic reporting

1. The structure of the "Introduction" should be rearranged in order to stress cannabidiol (CBD), the molecul of the scope, as promissing candidate for clinical utilization and thus it is of high importance to know whether CBD influences the expressionof P-gp and/or BCRP.
2. Authors should explain in introductory part if there is chance to use CBD therapeutically in pregnancy.
3. Page 2, line 13, word polarizer shoud be changed to polarized
4. Source of cells should be given more precisely, and the studies, in which the cells were used should be refered, since various clones of MCF-7 and BeWo cells exist differing in P-gp and/or BCRP expression (e.g. Ceckova et al. 2006)
4. Page 4, line 12, citations Golan et al 2009 and Feinshtein et al 2010 are provided to describe cell cultivation. These citation should be reconsider as only the latter deals with JAr cell lines. MCF-7 and Bewo cultivation is not described.
5. page 5, line 25, name of the company producing High capacity cDNA RT kit should be provided
6. Word chronic when speaking about exposure should be omitted from the text of the manuscript

Experimental design

1. PCR analysis should be remade, using one housekeeping gene for the analysis is not sufficient to achieve desired accuracy, especially in this study, where differences in mRNA expresssion are 2-2,5 fold at maximum.
2. Results desribing expression of P-gp/BCRP in all cell lines used is incosistent. Immunocytochemistry of BCRP should be added (figure 1)
3. Description of the results from protein level to mRNA is questionable. Authors should reconsider this scheme and to think about semi-quantitative analysis of protein expression usin immunohistochemistry instead of qRT-PCR as mRNA analysis seems as in this context as a step back

Validity of the findings

1. Figure 2 - the results provided seem to be questionable. Based on the analysis of actin or Na+/K+ ATPase it is obvious that protein load in gel was not the same everywhere, moreover, intensity of bands of these proteins correlates with intensity of bands of target proteins, indicating that the changes in protein expression observed are not correct. Does the software used EZQuant-Gel 2.11 normalize the target protein amount to reference protein properly? Moreover, in e.g. figure A for 72 h, it seems unlikely that 1.25 fold increase with SD +- 20% has P equal to **, especially when one-way ANOVA is used for three replicates. So, how many biological replicates were used as source data for analysis, it should be stated precisely. Similarly in figure B, decrease in BCRP protein expression for 72 h in BeWo should be discussed
2. Figure 3 (B,D), the wester blots as provided cannot be used for semi-quantitative analysis, the results as shown seem to be irrelevant.
3. Figure 4, picture of western blot is completely lacking
4. Figure 5, again authors must state how many samples (it means biological replicates) have been used for the analysis and the same number of samples must be used for all calculations. This figure shows the higher SD the better statistical significancy despite the same increase (2 fold), moreover, the changes observed can be the result of instability of the housekeeping used. This must be corrected. Comments concerning figure 2-4 raise dobts about validity of the study outcomes.
5. P-gp and BCRP are considered to have similar transcriptional regulation, authors should discuss the opposing effect of CBD on P-gp and BCRP.

Reviewer 2 ·

Basic reporting

This manuscript is clearly written and conforms to professional standards. The background information provided in the introduction and discussion is excellent. The structure of the article conforms the journal style, and the figures are nicely composed and labeled.

Experimental design

The research question is clearly stated, the experiments are well designed, the methods are well described, and the investigations have been conducted rigorously.

Validity of the findings

The data is strong and is presented in a clear and well understandable way. The results make sense in the context of the research hypothesis. The conclusions are appropriate for the findings.

Additional comments

This is an excellent study which aimed to investigate the effect of cannabidiol on the expression of ABC transporters P-gp and BCRP in trophoblast-like cells. This is important since cannabis use is an increasing problem during pregnancy, and its potential consequences on the placenta and the fetus are of major concern. The manuscript is insightful and very well written, the statistics and the representation of the data are proper, and the paper is easily readable. Therefore, I suggest to have it accepted for publication, and I have only minor comments to be addressed before publication:

1) In several places all along the text some periods are misplaced. This needs to be fixed.
2) Page 2: Please write “polarized” instead of “polarizer”.
3) Please rephrase the sentence on Page 2, line 12, since the word “constructed” does not seem to be the best fit.
4) Page 2, line 20: Please write “protein” after kDa.
5) I would write “trophoblast-like” instead of “trophoblast” cell lines, since JAR and BeWo cells are choriocarcinoma cell lines.
6) On Page 4, I would put “h”: after 24 and 48.
7) On Page 5: Please describe the antibodies used for the Na/K ATPase and NFkB Western blots.
8) DAPI (4',6-diamidino-2-phenylindole) is written with capital letters. Please define first and write this way.
9) Please provide vendor information for DAPI.
10) On Page 6: correctly it is called Delta-Delta Ct method.
11) Part of the statistical analysis is missing, which needs to be added.
12) Page 8, lines 1 and 110: The use of “placenta” is not correct in this context.
13) It would be useful if subheadings could be used in the Discussion.
14) Reference is missing from the sentence ending on Page 10/line 13,
15) Figure legends are uploaded twice.
16) Figure 1: Please define “NC”.

Reviewer 3 ·

Basic reporting

No comments

Experimental design

The study is very preliminary and completely descriptive. I have several concerns regarding the experimental design and the interpretations of the results.

Validity of the findings

No comments

Additional comments

In this study the authors have investigated the effect of cannabidiol on two cell lines derived from human choriocarcinoma. The conclusions are based on descriptive data of mRNA and protein abundance. Nevertheless the results are discussed as if the changes observed would affect the placental barrier.

Major points:

1. Previous studies have already shown the expression in BeWo and JAr cell lines of a complete panel of transporters and pumps, in addition to MDR1 and BCRP (see for instance Serrano et al., Placenta 28 (2007) 107-117). Several results described here as original (for instance page 7, lines 6-8) should be commented as mere confirmation of previous reports.

2. Page 8, line 9. The present paper offer little clue on the mechanism of action of cannabidiol on the expression of these two ABC proteins. The fact that mRNA levels were consistent with protein abundance is an interesting data but adds little to the mechanism controlling the expression of these genes.

3. Are the observed effects due to specific or non-specific (for instance chemical stress-mediated) mechanisms? Are cannabinoid receptors expressed in JAr and BeWo cells? Are they responsive to cannabidiol? What is the intracellular signalling pathway involved? Is there any evidence for a role of any nuclear receptor?

4. Page 11, lines 15-22. This paragraph is highly speculative. Are the authors proposing an autocrine mechanism? What are the meadiators?

5. Page 12, lines 8-13. The conclusion is again very speculative, above all, the last sentence.

6. Figure 1. Why BCRP was not included in this part of the study?

7. Figure 1. Immunolocalization of MDR1 does not seem to be at the plasma membrane of BeWo and JAr cells, but intracellular. Is BCRP also located intracellularly? How this fit the results from the western blot analyses?

8. Does the treatment with cannabidiol change the subcellular localization of these proteins?

9. Any functional study would be required to suggest that cannabidiol is able to change the role of these proteins in the placental barrier.

---

## Round 0.2 · accepted · Accept

The revision addressed all the criticism raised by the reviewers.